# Long-Term Brain Disorders in Post Covid-19 Neurological Syndrome (PCNS) Patient

**DOI:** 10.3390/brainsci11040454

**Published:** 2021-04-02

**Authors:** Domenico Nuzzo, Gaetano Cambula, Ignazio Bacile, Manfredi Rizzo, Massimo Galia, Paola Mangiapane, Pasquale Picone, Daniela Giacomazza, Luca Scalisi

**Affiliations:** 1Consiglio Nazionale delle Ricerche, Istituto per la Ricerca e l’Innovazione Biomedica (CNR-IRIB), 90146 Palermo, Italy; pasquale.picone@cnr.it; 2Dipartmento of Scienze Biologiche, Chimiche, Farmaceutiche e Tecnologiche (STEBICEF), University of Palermo, 90133 Palermo, Italy; 3Unità Operativa Complessa Radiologia P.O. S. Antonio Abate—Azienda Sanitaria Provinciale di Trapani, 91100 Trapani, Italy; gaetanocambula@libero.it (G.C.); ibacile56@gmail.com (I.B.); 4Department of Health Promotion, Mother and Child Care, Internal Medicine and Medical Specialties, University of Palermo, 90133 Palermo, Italy; manfredi.rizzo@unipa.it; 5Dipartimento di Biomedicina, Neuroscienze e Diagnostica Avanzata, Università degli studi di Palermo, Via del Vespro 127, 90133 Palermo, Italy; massimo.galia@unipa.it; 6Clinica Morana, 91025 Marsala, Italy; paola.mangiapane@virgilio.it; 7Consiglio Nazionale delle Ricerche, Istituto di Biofisica (CNR-IBF), 90146 Palermo, Italy; daniela.giacomazza@cnr.it; 8Centro Medico di Fisioterapia “Villa Sarina”, 91011 Alcamo, Italy; 9Azienda Sanitaria Provinciale Di Trapani (ASP 9 TP), 91100 Trapani, Italy

**Keywords:** post-COVID-19, neurological disorders, SARS-CoV-2, brain damage, COVID-19

## Abstract

In the recent pandemic disease, called COVID-19, the role of neurologists and neurobiologists represents a chance to study key features of brain infection and deepen neurological manifestations of COVID-19 and other coronavirus infections. In fact, many studies suggest brain damage during infection and persistent neurological symptoms after COVID-19 infection. Reverse transcription PCR test, antibody tests, Computed Tomography (CT) of the lung, and Magnetic Resonance (MR) of the brain of the patient were periodically performed during this case report for eight months after infection. The aim of this article is to describe the prolonged neurological clinical consequences related to COVID-19. We believe it is clinically clear that we can define a post-acute COVID-19 neurological syndrome. Therefore, in patients after a severe clinical condition of COVID-19, a deepening of persistent neurological signs is necessary.

## 1. Introduction

Coronaviruses (CoVs) are large positive-stranded enveloped RNA viruses that generally cause respiratory diseases in humans and in animals. A highly pathogenic CoV, named SARS-CoV-2 or COVID-19, dramatically emerged in December 2019 in Wuhan, China [1]. This new CoV rapidly spread around the world, inducing a pandemic state that is firstly a serious health emergency, but also a devastating social-economic problem. The disease manifestations are not limited to the respiratory system but other organs can be affected. In particular, virus-related neurological manifestations are being reported more frequently in the scientific literature [2]. Moreover, symptoms in patients after acute COVID-19 are persistent for a long time. In fact, several studies indicate that many patients present symptoms even for a prolonged time after SARS-CoV-2 infection, indicating that attention towards these patients should be maintained [3]. 

Post-COVID clinical manifestations. About one third of positive patients develop neurological and neuropsychiatric symptoms, generally in the early stages of the disease, but sometimes even after the resolution of the respiratory symptoms [4]. The most common symptoms appearing post-infection are anosmia, ageusia or dysgeusia, headache, muscle and joint pain, fatigue and mental fog, symptoms that can last for weeks or months [4]. In severe cases, the infection can also lead to delirium and psychosis, inflammatory syndromes (such as encephalitis and acute disseminated encephalomyelitis), ischemic and hemorrhagic strokes [4,5,6,7]. Many causes are at the basis of the neurological manifestations of COVID-19 being a direct effect of the coronavirus on the nervous system or immune-mediated effects linked to para-infectious or post-infectious mechanisms [7,8,9,10].

It has been found in several post-COVID-19 follow-up cases that together with a global weakness related to loss of muscle mass, 16% of patients present with disabling focal neurological deficits relating to multiple axonal mononeuropathies [11].

Indirect effects. Ischemic stroke, one of the main neurological manifestations of COVID-19, for example, is one of the systemic effects of the disease. It is mainly linked to the activation of the inflammatory cascade due to the presence of the viral infection followed by the so-called “cytokine storm”, that is an exaggeratedly violent reaction of the immune defenses which, instead of protecting from the virus, begin to attack the healthy cells of other organs. This results in systemic impairment, especially in the most severe cases of infection [12]. It has been proposed that viral infection can directly cause damage of the endothelium covering the internal surface of blood vessels, thus increasing the pro-thrombotic milieu through the production of thrombin and immune-mediated activation of platelets. Other factors, such as dehydration and infection-induced cardiac arrhythmias, may also play a role in determining acute ischemic stroke in COVID-19 [13]. 

In other cases, encephalitis, appearing in about 10% of patients affected by COVID-19 [14], can occur after treatment with immunosuppressive drugs or immunoglobulins and plasmapheresis. Serial clinical systemic immune inflammation indices have been used to provide extra useful information on the Post Covid-19 Neurological Syndrome (PCNS) [15,16,17].

Direct effects. In some cases, encephalitis can directly originate from infection of the brain by the Coronavirus, as demonstrated by the presence of viral RNA and the production of specific virus antibodies in the cerebrospinal fluid [18]. This direct effect can be explained considering that the Ace2 receptor is the door by which enter human cells. Penetrating through the olfactory nerves directly into the brain, SARS-CoV-2 can reach the Ace2 receptors present on the neurons, thus establishing a link between the Spike protein and the receptor [9,19]. Recent studies have shown that neuropilin-1, a membrane protein highly expressed in neurons, is a factor facilitating the entry of SARS-CoV-2 into the cells of the nervous system [20]. The manifestations of encephalopathy are extremely variable in terms of severity, ranging from the simple headache to cognitive disorders, including mental confusion, delirium or dementia [21]. The encephalopathy can be a consequence of SARS-CoV-2 infection, especially in elderly people having pre-existing chronic diseases [19]. Patients with pre-existing cognitive decline or dementia, for example, may present a worsening of these pathologies. Patients over 60 years of age predisposed to cerebrovascular diseases, arterial hypertension, diabetes or dyslipidemia, seem to present a higher risk of ischemic stroke during COVID-19 [22].

Peripheral nervous system involvement. The involvement of the peripheral nervous system can occur in the form of acute neuropathies or polyneuropathies, the so-called Guillain-Barré syndrome. It is an acute inflammation of the nerves caused by an attack to the proteins present on the nerve sheath (myelin) by the antibodies produced by the patient’s immune system to fight the SARS-CoV-2. These antibodies in some cases, can assault the structures of the nervous system through a mechanism of molecular mimicry. The antibodies produced by the patient’s immune system against the virus “get confused” and attack the myelin layer of the nerves, causing nerve damage resulting in loss of strength muscles. In 15% of cases of Guillain-Barré syndrome, the respiratory muscles are affected, thus the mechanical ventilation is needed. Even though Guillain-Barré syndrome is a rare complication of COVID-19, it requires careful vigilance for early diagnosis and prompt treatment, due to the serious respiratory complications [23].

## 2. Case

### 2.1. Hospital Phase

A previously healthy 56-year-old man presented to the hospital with 5 days of fever and dyspnea. Chest Computed Tomography (CT) revealed in all lung lobes multifocal and peripheral areas of interstitial thickening with a “ground glass” appearance due to interstitial pneumonia (Figure 1).

Laboratory results showed high levels of inflammatory markers (see Appendix A), the SARS-CoV-2 reverse transcription PCR test and the SARS-CoV-2 antibody test were positive (March 2020). The patient developed a worsening of respiratory status in the next 24 h after hospitalization, requiring respiratory support and intensive care; in this context brain-CT was not performed during the following days. Magnetic Resonance Imaging (MRI) of the brain was performed and showed hyperintensity signal of cerebral-spinal fluid, especially in the left frontal lobe, in the fluid-attenuated inversion recovery (FLAIR) images (Figure 2).

After one month of intensive care (April 2020), the patient was transferred to sub intensive care, due to the improvement of the respiratory functions. The neurological examination indicated that the patient showed difficulty walking, weakness in the lower limbs, lack of strength in the pelvic girdle muscles and skin hyperalgesia especially in the backside. Chest CT showed almost complete resolution of the pulmonary damage (Figure 3). In May, SARS-CoV-2 reverse transcription PCR test and antibody test were negative.

### 2.2. Post Hospital Phase

At home, the patient was drowsy and lethargic. The neurological examination showed diffuse hypotonia and significant weakness in all four limbs. Absence of movement due to prolonged bed rest and immobilization caused a reduction in muscle mass: −30% in 2 weeks (Table 1) and joint stiffness. Deep tendon reflexes were diminished overall. The patient could hardly take a few steps with the help of a support walker and he was able to perform bed mobility tasks (from bed to chair, from sitting to standing position) with assistance. The patient reported that smell and taste were not yet recovered. 

In June 2020, three months after the positive results to COVID-19, the patient complained of chronic fatigue, headache, finger paresthesia, anxiety attacks and profound depression. The patient was subjected to brain MRI with intravenous paramagnetic contrast medium injection. In long TR images, numerous hyperintense focal areas were detected in the periventricular and subcortical white matter and in semioval centers. These imaging findings were referred to gliotic outcomes on a microvascular basis from a probable previous vasculitis episode. There were neither lesions of acute microischemic significance nor areas of pathological enhancement after paramagnetic contrast medium administration (Figure 4). 

After 5 months from negativization (August 2020), the patient continued to present neurological disorders together with a deep depression state. This condition further contributed to the unclear question about the involvement of the central nervous system in the COVID-19 infections. The EMG of the upper limbs has been done. The potential of the motor unit showed an increased amplitude and tracing striving of intermediate type. No denervation signs were detected in the checked muscle areas. The nerve conduction was normal.

At the moment of the drafting of the present article (September 2020), the patient is still suffering from short-duration epileptic seizures requiring the EEG periodical examination. Echocardiography and Transcranial Doppler did not show any cardiac causes of cerebral micro-embolism.

## 3. Discussion and Conclusions

Several literature reports have already shown the SARS-CoV-2 neurotropism; although the etiopathology of neuronal damage is not entirely clear [24]. Regarding the follow-up and the rehabilitation process of patients previously affected by SARS-CoV-2, some articles have been published with data and pathways still incomplete and to be designed. The alterations of the cerebral parenchyma found and documented in our patient, with the recent study of MRI, are findings completely similar to those that are recognized in about 40–45% of patients suffering from migraine to system immune disorders, as well as inflammation of the connective tissue. These conditions, which have as their common denominator a vascular microsuffering caused by vasoconstriction, generate microthrombosis with consequent microvascular alterations, leading to neuronal degeneration and subsequent gliosis. These areas evolve in minute focal areas of altered signal intensity recognizable in neuroimaging with Magnetic Resonance as signal hyperintensity in long TR images, especially in FLAIR, expression of gliosis reactive to neuronal damage. These additional aspects of COVID-19 infection can further evidence the harmful action of COVID-19 in the central nervous system.

In the clinical-instrumental management of post-COVID patients, an MRI control of the brain should be provided even in basic conditions (wide accessibility exam), in order to document any alterations of the brain parenchyma to better define post-COVID-19 neurological syndrome. Therefore, patients affected by severe COVID-19 need a multidisciplinary team to address it.

## Figures and Tables

**Figure 1 brainsci-11-00454-f001:**
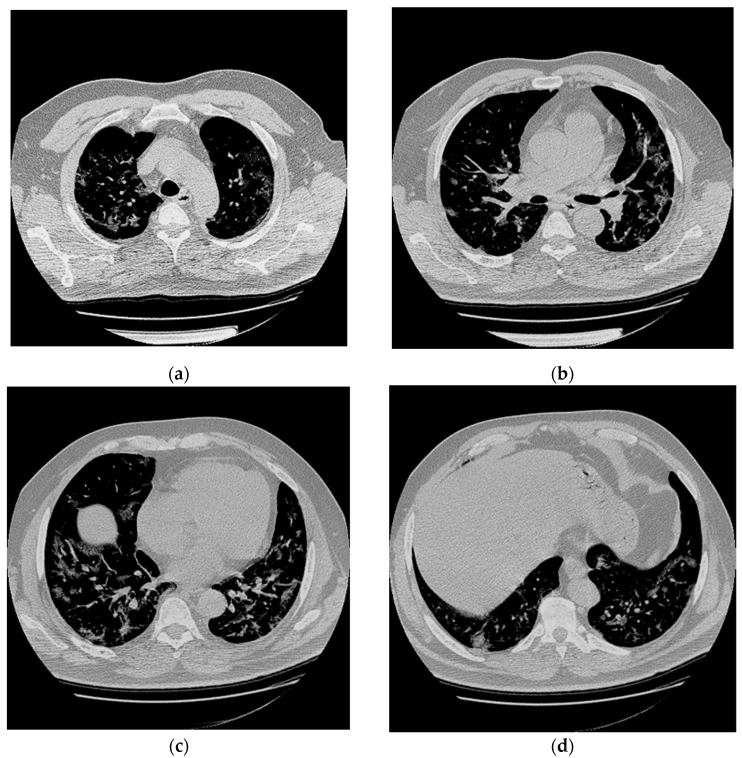
Chest CT abnormalities in a 56-year-old man with positive RT-PCR test results for SARS-CoV-2. (**a**–**d**) Axial nonenhanced chest CT images (lung window) show diffuse bilateral pulmonary ground-glass opacities and dilated segmental and subsegmental vessels.

**Figure 2 brainsci-11-00454-f002:**
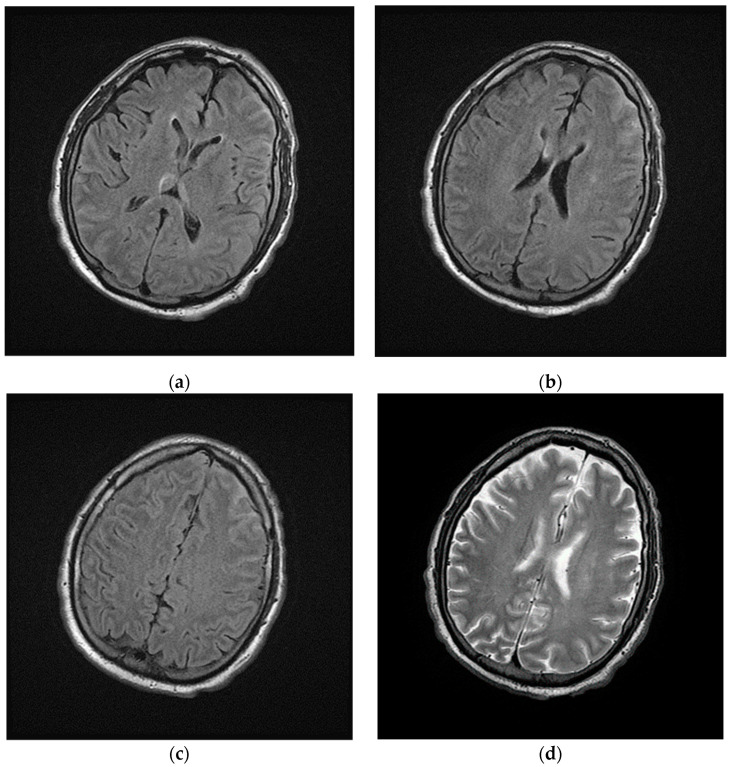
Magnetic Resonance Imaging (MRI) of brain performed during intensive care. (**a**–**c**) Axial fluid-attenuated inversion recovery (FLAIR) images and (**d**) axial T2w image show cerebrospinal fluid hyperintensities in the left frontal lobe; in the same images no focal lesions of the cerebral parenchyma were detected.

**Figure 3 brainsci-11-00454-f003:**
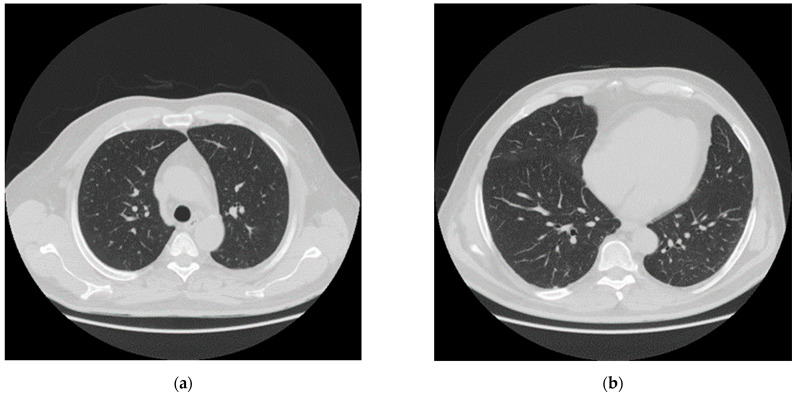
CT after one month of intensive care and one month of sub intensive care. (**a**,**b**) Axial nonenhanced chest CT images (lung window) show almost complete absorbent of lesions.

**Figure 4 brainsci-11-00454-f004:**
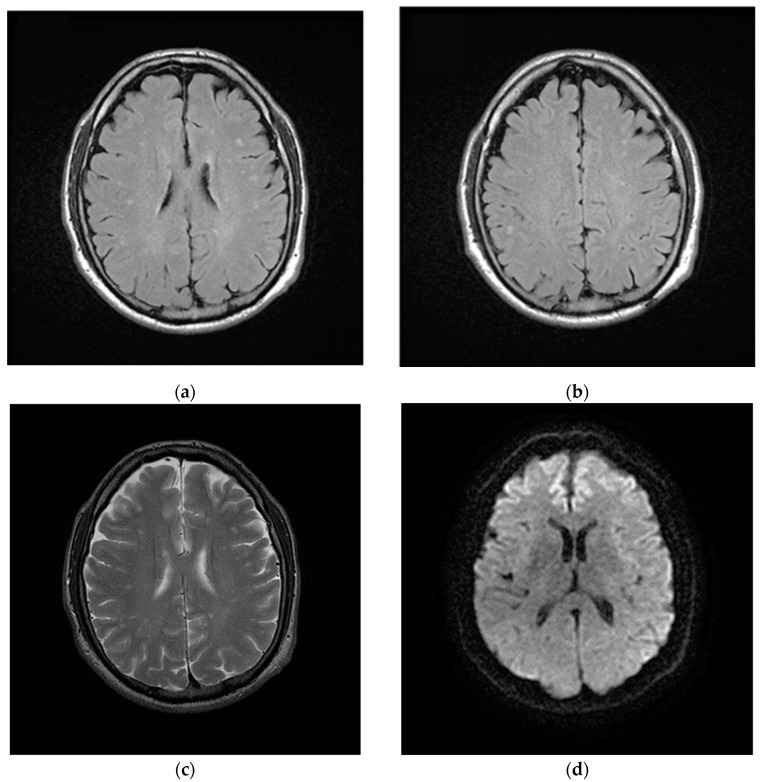
Magnetic Resonance Imaging (MRI) of brain after discharged from the hospital. (**a**,**b**) Axial FLAIR and (**c**) axial T2w images show nonconfluent multifocal white matter hyperintense lesions; (**d**) On axial diffusion weighted image (DWI) the same lesions do not show impeded diffusion. These lesions were referred to gliotic outcomes.

**Table 1 brainsci-11-00454-t001:** Anthropometric measure from pre to post infection. Abbreviations: 6-Minute Walking Test (6-MWT); Body Mass Index (BMI).

Characteristics	Pre Infection December	After Intensive Care April	Post Infection September
Weight (Kg)	104	75.8	81.1
BMI	34	25.7	28.1
Fat (%)	37.4	24.8	30.4
Status	Obese	Normal weight	Overweight
Diseases	No	…	Neurological disorders
Psychiatric activity	Normal	…	Depression
6-MWT (meter)	620	<1	510
Chest (cm)	128	105	107
Waist (cm)	141	103	104
Biceps (cm)	31	25	27
Leg (cm)	51	45	47

## Data Availability

The subject gave his informed consent for inclusion before his/her participation in the study. The study was conducted in accordance with the Declaration of Helsinki. The request for approval has been submitted and due to the vaccination campaign currently underway in our country the response suffered a slight delay. It will be sent on the next 26 April.

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
