# Peer review of "Long-Term Brain Disorders in Post Covid-19 Neurological Syndrome (PCNS) Patient"

_brainsci, 2021, doi:10.3390/brainsci11040454_

Round 1

Reviewer 1 Report

This is an interesting case report, however further detailed information is needed to draw a useful conclusion for the reader. 

Electrophysiology data must be added as well as clinical examination of the patient to differentiate between central and peripheral nerve disorders (reflexes, pyramidal signs, etc?)

Multifocal neuritis has been described (Neeham et al) and should be cited and discussed. 

The MRI is suggestive for changes associated with SARSCOV infection, however, not clearly  for the clinical presentation described

Author Response

First of all, we would like to thank the referee for his/her revision.

This is an interesting case report, however further detailed information is needed to draw a useful conclusion for the reader.

Electrophysiology data must be added as well as clinical examination of the patient to differentiate between central and peripheral nerve disorders (reflexes, pyramidal signs, etc?)

  • The EMG of the upper limbs has been done. The potential of the motor unit showed an in-creased amplitude and tracing striving of intermediate type. No denervation signs were detected in the checked muscle areas. The nerve conduction was normal.
  • Response asymmetry of the sensory motor activity of the ulnar nerve with right minus with respect the left one. The amplitude of the sensory motor activity was comparable the previous one. Absence of conduction interruption buy examining the right ulnar nerve. Normal potential of the motor unit of the bilateral axillary nerve.  Normal sensory and motor evoked potentials in the four limbs.

Conclusions: Chronic left radiculopathy C5-C6. Mild neuropathy of the left sensory ulnar nerve.

A summary of what described above has been inserted in the manuscript 

Multifocal neuritis has been described (Needham et al.) and should be cited and discussed.

A brief comment about the occurrence of multifocal neuritis as be inserted in the text.

The MRI is suggestive for changes associated with SARSCOV infection, however, not clearly for the clinical presentation described

The clinical presentation has been changed.

To complete the patient condition, we would like to add that the evidence explaining the epileptic episode is the congenital asymmetry of the right side of the hippocampus with accentuation of the parahippocampal cistern. We hypothesize that the COVID-19 or the associated therapy have lowered the epileptogenic threshold of the patient.

Reviewer 2 Report

The authors present a case report of a patient with prolonged hospitalization. secondary to COVID-19 and having post-COVID neurologic syndrome. There is very little detail into the patient's hospital course, not including regular laboratory values. It does not appear the patient had cerebrospinal fluid sampling, and only MR imaging. 

It is important to note that we are not routinely getting MR brain imaging on other patients with severe prolonged critical illness, and this could be a phenomenon of critical illness and not COVID-19 at all. 

The authors do not prove causality nor do they give the full details of the patient's clinical course. 

MR brain imaging findings in COVID patients has been previously described and is not of major interest. 

Author Response

We would to thanks the Referee for the interesting notes.

The authors present a case report of a patient with prolonged hospitalization. secondary to COVID-19 and having post-COVID neurologic syndrome. There is very little detail into the patient's hospital course, not including regular laboratory values. It does not appear the patient had cerebrospinal fluid sampling, and only MR imaging.

More details and laboratory data recorded during the hospitalization phase are now been added in supplementary data. Cerebrospinal fluid was not collected. In fact, during the pandemic emergency, only chest CTs were performed in patients with severe SARS-CoV-2 infection. Brain involvement is a recently discovery as a consequence of SARS-CoV-2 infection.

It is important to note that we are not routinely getting MR brain imaging on other patients with severe prolonged critical illness, and this could be a phenomenon of critical illness and not COVID-19 at all.

Thanks for the observation. We hypothesize that severe SARS-CoV-2 infection along with intensive care and use of respiratory support could result in intense central damage. Several scientific reports indicate the presence of neuronal signs due to SARS-CoV-2 infection.

It is important to highlight that before infection the subject was healthy and had no evidence of neurological disorders or other diseases.

The authors do not prove causality nor do they give the full details of the patient's clinical course.

MR brain imaging findings in COVID patients has been previously described and is not of major interest.

This case report describes a long neurological manifestation started after COVID-19. The patient presents, until today, neurological symptoms. We believe this is important to define the clinical intervention of long post-acute COVID-19 neurological syndrome.

Reviewer 3 Report

This is an interesting case worth reporting. The manuscript needs subtantial revisions. Please review the published work on Post Covid-19 Neurological Syndrome which is reasonably well described as the references listed here.

  1. Read them well. Note the very well described pathobiology of the disease and strong similarity with the pathobiology of stroke.
  2. Re-write the abtract- this should be straight to the point stating you are describing a case of PCNS ( is this the first such case in your country?
  3. Give us a decription of serial changes of neutrophil-lymphocyte ratio and changes in serial systemic immune inflammatory indices (SSIIi, Wijeratne & Wijeratne 2021), if you can access them.
  4. Introduction should be short- no need to talk about the pandemic status which is very well known to our readers.
  5. Describe the case the clical course as short as possible and expand the discussion further

suggested references;

  1. Wijeratne T, Wijeratne C. Clinical utility of serial systemic immune inflammation indices (SSIIi) in the context of post covid-19 neurological syndrome (PCNS). J Neurol Sci. 2021 Feb 20;423:117356. doi: 10.1016/j.jns.2021.117356. Epub ahead of print. PMID: 33636659; PMCID: PMC7895697.
  2. Anrather J., Iadecola C. Inflammation and stroke: an overview. Neurotherapeutics. 2016;13(4):661–670. [PMC free article] [PubMed] [Google Scholar]
  3. Iadecola C., Anrather J. The immunology of stroke: from mechanisms to translation. Nat. Med. 2011;17(7):796–808. [PMC free article] [PubMed] [Google Scholar]
  4. Iadecola C., Anrather J., Kamel H. Effects of COVID-19 on the nervous system. Cell. 2020;183(1) 16–27.e1. [PMC free article] [PubMed] [Google Scholar]

Author Response

Thank you very much for having commented the manuscript. We believe your indications have improved our case report.

This is an interesting case worth reporting. The manuscript needs substantial revisions. Please review the published work on Post Covid-19 Neurological Syndrome which is reasonably well described as the references listed here.

  • Read them well. Note the very well described pathobiology of the disease and strong similarity with the pathobiology of stroke.

The alterations found in the MRI can be related to a microcirculation damage as can be observed in the chronic microvascular leukoencephalopathies, autoimmune pathologies, vasculitis, and migraine. The ischemic stroke consists in a single acute event related to a specific vascular area, strictly near the damaged vessel. These differences let us lean toward a different neuronal damage. 

  • Re-write the abtract. This should be straight to the point stating you are describing a case of PCNS (is this the first such case in your country?

Thank you for this suggestion. The abstract has been modified. This is the first long-COVID-19 case reported in our country and involving brain damage.

  • Give us a descriptions of serial changes of neutrophil-lymphocyte ratio and changes in serial systemic immune inflammatory indices (SSIIi, Wijeratne & Wijeratne 2021), if you can access them.

We agree with the referee that the knowledge of these parameters could be useful to describe the patient condition, but, unfortunately, we do not have access to this information.

  • Introduction should be short- no need to talk about the pandemic status which is very well known to our readers.

In agreement with referee requirement and to the guideline of the journal the introduction has been modified.

  • Describe the case the clinical course as short as possible and expand the discussion further

Required changes have been done.

We would like to thank the referee for the suggested references, now introduced in the manuscript.

Round 2

Reviewer 3 Report

Please use Post Covid-19 Neurological Syndrome (PCNS) rather than post COVID019 Syndrome  ( line 75, reference 15). Please add the reference 1.Wijeratne T, Crewther S. Post-COVID 19 Neurological Syndrome (PCNS); a novel syndrome with challenges for the global neurology community. J Neurol Sci. 2020 Dec 15;419:117179. doi: 10.1016/j.jns.2020.117179. Epub 2020 Oct 13. PMID: 33070003; PMCID: PMC7550857.

2. Wijeratne T, Crewther S. COVID-19 and long-term neurological problems: Challenges ahead with Post-COVID-19 Neurological Syndrome. Aust J Gen Pract. 2021 Jan 12;50. doi: 10.31128/AJGP-COVID-43. Epub ahead of print. PMID: 33543150.

before reference 15. These two papers decribe the clinical features of PCNS well. 

Author Response

Dear Reviewer,

The suggested references have been added as requested. Thanks again for your suggestions.